# An Overview of Naturally Occurring Retirement Communities (NORCs) for Ageing in Place

Jiaxuan E [1], Bo Xia [1,*], Connie Susilawati [2], Qing Chen [1] and Xuechun Wang [1]

1   School of Architecture and Built Environment, Faculty of Engineering, Queensland University of Technology, 2 George Street, Brisbane, Qld 4120, Australia; jiaxuan.e@hdr.qut.edu.au (J.E.); richard.q.ch@gmail.com (Q.C.); xuechun.wang@hdr.qut.edu.au (X.W.)
2   School of Economics and Finance, Faculty of Business and Law, Queensland University of Technology, 2 George Street, Brisbane, Qld 4120, Australia; c.susilawati@qut.edu.au
*   Correspondence: paul.xia@qut.edu.au

**Abstract:** As an alternative to ageing at home in the community, naturally occurring retirement communities (NORCs) have great potential to facilitate ageing in place; however, they have not attracted much research attention. This study conducts an overview of NORCs, aiming to examine the previous research in a comprehensive manner in order to explore how NORCs impact ageing in place, with the goal of guiding future research. The research presented here employs the content analysis method to review prior NORC-related studies and categorise research themes and findings following top-down coding principles. A total of 49 articles were selected from the Scopus and Web of Science databases, and the results show that the "social environment", which was the most discussed topic (n = 24), provides the necessary mental support and physical motivation for older adults to live actively in NORCs, and that NORCs play a positive role in preserving public resources and promoting individual health. The limitations of this study include the fact that there is little public information on NORC programs and the subjective classification of themes, among others. This study acts as a foundation for future research on NORCs, which serve as a perfect model for healthy ageing in place.

**Keywords:** ageing in place; naturally occurring retirement community; NORC; NORC-SSP; social environment; built environment; wellbeing; older resident



## 1. Introduction

The world's population is ageing. According to the United Nations [1], at the global level, approximately 9% of people were aged 65 or over in 2019, and the proportion is expected to reach nearly 12% in 2030, 16% in 2050, and 23% by 2100. Especially in developed counties, the ageing populations are an established trend. For example, in 2019, 18% of the population in Europe and North America was 65 and older, followed by 16% in Australia and New Zealand. According to current projections, one out of four Europeans and North Americans could be 65 or older by 2050. As one of the most significant social changes of the twenty-first century, the ageing of the population will affect nearly every sector of society, including the labour and financial markets, housing, transportation, social security, as well as family structures and intergenerational relationships [2].

Along with the established trend of population ageing, most older people prefer to age in place, which means staying at home in the community. In particular, with the baby boomer cohort having reached 60 years of age in 2006, many of those who were born from the mid-940s to the mid-1960s are now fuelling a growing demand for ageing in place [3] and the corresponding support and care required. Unfortunately, ageing in place is often challenging for older adults because of a variety of common barriers, including diminished physical abilities, the rising costs of long term care, the increasing risk of social isolation, and a lack of preparedness in the community environment [4]. In particular, the effects

of social isolation and loneliness on the health and wellbeing of older people have been demonstrated [5].

In order to better facilitate ageing in place, naturally occurring retirement communities (NORCs), a concept that originated in the United States of America in the 1980s, have emerged as a model of collaborative care that can support older people, allowing them to remain in their homes as long as possible and avoid a shift to more restrictive environments [6]. A NORC is a neighbourhood or building complex that was not originally designed for older adults but eventually came to accommodate a large percentage of older residents. While not initially created to help older adults age in a community, NORCs have evolved naturally and provide a way for older adults to live independently [7]. Furthermore, NORC supportive service programs (NORC-SSPs) have been implemented and developed for more than 30 years; they have improved the physical and psychological health of the older participants, increased the efficiency of resource allocation from service providers, enhanced funding support from government and related organisations, and promoted the establishment of more beneficial ageing policies by policy makers [8,9].

Despite the great potential of NORCs to facilitate ageing in place, they have not attracted much research attention. On the one hand, although NORCs are already a factual phenomenon globally, there is limited research on NORCs, with most of the existing studies conducted in the U.S., where NORCs originated. On the other hand, the majority of prior studies concentrated on one specific area of interest, such as the social network of older adults, the relationship between service providers and older residents, and older individuals' participation in program activities. What is lacking is a comprehensive review of previous NORC studies in order to form a holistic picture of how NORCs support successful ageing in place.

Therefore, this study attempts to fill the gap in the literature and is the first to conduct an overview of the research conducted on naturally occurring retirement communities since 1986, when the concept of a NORC was first presented. This study aims to examine the current status of the research on NORC development and evolvement, explore the impact of NORCs on ageing in place, and identify future directions for NORC research.

## 2. Literature Background

### 2.1. Ageing in Place

Ageing in place is defined as "the ability to live in one's own home and community safely, independently, and comfortably, regardless of age, income, or ability level" [10]. As independent living may be a better option for the older population and for society compared to assisted living, such as nursing homes [11], the aim of many ageing-in-place programs is to enable older persons to remain independent in communities where social networks of family and friends have been established.

It has been demonstrated that ageing in place leads to many positive outcomes. The social sector offers opportunities for preventing premature institutionalisation, delaying the demand for costly health services, creating efficiencies of scale in service delivery, and increasing possibilities for community involvement, volunteerism, and leadership [12]. At the individual level, retirement in place can improve self-efficacy, provide social support within the community, and allow you to maintain a sense of familiarity and belonging [13]. Furthermore, cognitive functions, daily life activities, and depression have also been reported as improved [14].

However, as individuals' lives change, so do the environment and policies that may affect their residence over time [15]. Lau and Scandrett [16] identified three types of barriers to ageing in place: individual, community, and social. The barriers at the individual level call for preventing diseases that may cause disabilities (e.g., limitations on daily living activities) [17], maintaining social ties (e.g., family and neighbourhood) [18], and modifying the home/living environment in a timely manner [19]. At the community level, services that cater to the social and health needs of vulnerable older people are essential [20]. Finally,

at the societal level, providing adequate resources to each community's older residents requires public assistance [21].

Similarly, Cutchin [22] identifies managing instability or irregular changes in the individual's circumstances and needs as a major obstacle to successful ageing in place. Individuals, local communities, and society must make concerted efforts to enable those with disabilities to perform necessary home renovations and live in their current home [16]. As a matter of fact, a 2011 study found that only 18% of households had lived in the same house for 20 years or more, despite their strong desire to do so [23]. Taking this into account, communities such as NORCs can leverage coordinated efforts among their members to facilitate ageing in place, and this deserves further study [24].

### 2.2. Definition of Naturally Occurring Retirement Community (NORC)

A naturally occurring retirement community (NORC) describes a community that was not designed for the needs of older people but has a significant proportion of senior residents due to natural migration patterns [25]. The attractions of NORCs include both neighbourhood services that support older people's needs and capabilities as well as safety and close proximity to age peers. Because older people are concentrated in geographically close areas, it is possible to serve them effectively and facilitate formal and informal cooperation among residents, communities, service providers, and the public sector. Therefore, in order to facilitate the physical and psychological wellbeing of older people, the NORC is viewed as a critical model for ageing [26].

However, very few studies have tried to clearly define the concept of a NORC. Usually, studies that define NORCs agree on the composition, but the details differ. It is generally believed that a NORC is a geographical area in which a large proportion of older residents live in a specific area or in housing that was not designed or planned for the older people at the beginning. However, what constitutes a "large proportion" of the population and at what age a person should be included in that proportion are not agreed upon.

Hunt and Gunter-Hunt [25] first defined the term "Naturally Occurring Retirement Community" as "a housing development that is not planned or designed for older people, but which over time comes to house largely older people". In addition, they mentioned that NORCs may vary considerably in scale. For example, NORCs can range from a local neighbourhood with a disproportionate number of older residents to an apartment building or complex. According to Hunt and Ross [27], a NORC is a type of housing development that does not plan or design for older people but has a significant proportion (over 50%) of residents at least 60 years old. Since apartments were the most common form of alternative housing for older people in the U.S. at the time, the authors focused on apartment NORCs, despite NORCs having many different forms.

Having at least 40% of household heads who are 65 or older in a census block group (for a total of at least 200 households) was defined as a NORC by Lanspery and Callahan [28]. Sixty-five was chosen as the demarcation line instead of 60, based on Hunter's recommendation, because it provides an estimate of NORCs that is more conservative, as 65 is the Medicare eligibility age. Lanspery and Callahan [28] set a minimum number, i.e., 200, rather than household proportions. They were concerned with the opportunities that NORCs provide for supportive services. The 200-household threshold represents the mid-range of the scale generally considered sufficient to support a full-time services coordinator in senior housing.

In New York, a NORC is defined as a region where at least 50% of households have a senior citizen or a housing complex with more than 2500 elderly residents [29]. The Atlanta consortium of local providers targeted NORCs as areas where 25% or more of the population is over 65 years of age in order to provide comprehensive service delivery to the population [13]. The consortium further determined that a census area with a higher proportion of people aged 75 and older living alone was a high-risk area. Lyons and Magai [30] defined a qualified NORC as having 65% or more residents aged 50 plus but did not explain their choice of housing community.

The U.S. Department of Health and Human Services issued a report that tried to identify the cut-off boundary on the age and number of older people in a NORC. They interviewed experts on ageing issues, and some people supported the use of 60 years old as the minimum for those who are considered seniors in order to be consistent with the Older Americans Act that defines the term "older individual" as an individual who is 60 years of age or older. However, others believed that the boundary should be determined by degree of disability rather than a specific age. The definition of "older" refers to people aged from 50 to 65 years old, while "significant proportion" is defined as 40% to 65% [31]. One survey by the American Association of Retired Persons in 2005 found that 36% of respondents (55 and older) lived in NORCs [32].

In 2006, the U.S. federal government specifically presented the definition of a NORC under Title IV of the Older Americans Act 1965 as "a community with a concentrated population of older individuals, which may include a residential building, a housing complex, an area (including a rural area) of single-family residences, or a neighbourhood composed of age-integrated housing where (i) 40% of the heads of households are older individuals; or (ii) a critical mass of older individuals exists, based on local factors that, taken in total, allow an organisation to achieve efficiencies in the provision of health and social services to older individuals living in the community, and that is not an institutional care or assisted living setting." [33].

With the development of geographic information system (GIS) and big data technology, Rivera-Hernandez and Yamashita [21] identified NORCs in the U.S. by GIS and employed the definition of 40% or more homeowners and renters aged 65 years and older. Due to the fact that older residents may need assistance or care regardless of whether they own a home or not, the numerator was homeowners or renters. Based on the availability of ABS census data and population scale in Australia, E and Xia [34] introduced an Australian version of the NORC that is defined as a community with 40% or more household members aged 65 years and older. The study used the concept of household members who usually reside in private dwellings as the basic unit to define NORCs, which excludes holiday visitors and persons who have moved to nursing homes.

Table 1 shows a variety of definitions of a NORC by different authors or organisations. It is important to distinguish between the two benchmarks, i.e., the proportion and number of older people for defining a NORC, because the proportion of older people helps to describe a community's character, while the number of older people has a greater impact on the implementation of supportive service programs. In densely populated urban areas, the proportion of the population meeting the selected age criteria may fall below the selected threshold and, therefore, not meet the NORC definition. In practice, however, the number of older adults may exceed the threshold where economies of scale could be realised. The concept of NORC supportive service programs (NORC-SSP, which will be discussed in detail in the following section) is frequently mentioned by some authors and experts when defining NORCs and sometimes used interchangeably. Separating the two concepts, however, has its benefits. NORCs are communities of people, some of whom may require services; NORC supportive service programs may be a valuable addition to such communities. There may be a large percentage of older adults in NORC communities who do not require supportive services. There may also be residents in other non-NORC communities who need supportive services.

**Table 1.** Definitions of Naturally Occurring Retirement Community.

| Definition | Researchers | Year |
| --- | --- | --- |
| A housing development that is not planned or designed for older people, but which over time comes to house largely older people | Hunt and Gunter-Hunt [25] | 1986 |
| Housing developments that are not planned or designed for older people but that attract a preponderance (over 50%) of residents at least 60 years of age | Hunt and Ross [27] | 1990 |

**Table 1.** *Cont.*

| Definition | Researchers | Year |
|---|---|---|
| In a census block group, at least 40% of the heads of households (for a total of at least 200 households) are aged 65 and over | Lanspery and Callahan [28] | 1994 |
| At least 50% of households have a senior citizen, or a housing complex with more than 2500 elderly residents | Yalowitz and Bassuk [29] | 1998 |
| The census block groups in which 25% of the population is over the age of 65. These communities can be considered Naturally Occurring Retirement Communities (NORCs) and considered for targeted comprehensive service delivery | Lawler [13] | 2001 |
| They qualified as NORCs because the building management provided limited to no formal social support and more than 65% of the residents were 50 years of age or older | Lyons and Magai [30] | 2001 |
| The age at which a person is considered "older" ranges from 50 to 65 years, and the definition of a "significant proportion" living in the community ranges from 40% to 65% | Ormond and Black [31] | 2004 |
| 36% of respondents 55 years and older could be viewed as living in NORCs | Kochera and Straight [32] | 2005 |
| A residential building, a housing complex, an area (including a rural area) of single-family residences, or a neighbourhood composed of age-integrated housing where 40% of the heads of households are older individuals | Senate and House of Representatives of the United States of America [33] | 2006 |
| 40% or more homeowners and renters aged 65 years and older | Rivera-Hernandez and Yamashita [21] | 2015 |
| A community with 40% or more members of households aged 65 years and older | E and Xia [34] | 2021 |

*2.3. NORC Supportive Service Programs (NORC—SSP)*

In the U.S., NORC programs, also known as NORC supportive service programs (NORC-SSPs), are designed to provide customised services to residents living in NORCs based on their specific needs. Residents, housing/neighbourhood associations, other community stakeholders, and health and social service providers collaborate on community-based programs. NORC programs may provide a variety of services, but all are aimed at providing older residents with optimal wellbeing and health so that they can take care of their independent living comfortably at home as they age.

Funding for NORC-SSPs is generally a mix of public and private contributions, which can include donations from charities, relevant government departments, private companies, community stakeholders, and residents and partners. A number of services are available through NORC-SSPs, including case management, transportation assistance, recreation and educational programs, and volunteer opportunities for older adults. A key feature of the NORC-SSP model is its ability to flexibly identify and deliver the types of services needed by older people ageing in place. The main factors that influence the supportive services programs are planning and design, staffing, marketing, program governance, program delivery and services, financing operations, lead agency, volunteer and intern activities, partnerships with other agencies, NORC management, NORC dynamics, and links to other resources [35].

The first NORC-SSP was established in New York City in 1986 at Penn South Houses, a ten-building cooperative housing development supported by the United Hospital Fund based on funding from United Jewish Appeal (UJA). More than 25 states across the country have replicated the NORC program model since then at local, state, and national levels.

Over 400,000 apartments in New York City, with most of the residents experiencing low and moderate income, were identified as potential NORCs in a 1991 study of housing occupancy rates [36]. In 1995, the state of New York passed legislation promoting the establishment of NORC-SSPs in low- and moderate-income housing developments in which at least half of the heads of household were aged 60 plus, or at least 2500 residents

were senior citizens. In 1999, New York City implemented the program but changed eligibility requirements to include developments with more than 250 older adults in which 45% of households have a head of household 60 years old or older or housing developments with more than 500 older adults. During the 2006 legislative session, the state legislature extended the program so that it would cover a NORC-SSP in low-rise neighbourhoods with fewer than 2000 older residents who do not share ownership [37]. By 2010, there were 54 NORC-SSPs running in New York State in housing developments and neighbourhoods with low and moderate income levels [38]. For years up to 2021, information is available for 33 NORC-SSPs in New York funded by the Department For The Aging (DFTA) [39].

Maclaren and Landsberg [35] discussed that NORC-SSPs in New York State have four objectives that make up their plan, as follows. First, provide effective and integrated community-based services that meet consumers' diverse needs. Second, improve preventive care and services that enable older people to live as independently as possible at home, thereby avoiding unnecessary long-term institutionalisation. Third, encourage consumers and their caretakers to take an active role in the key decisions that affect their care. Finally, deliver service quality and facilitate care by leveraging the unique characteristics of NORCs, such as the number and density of older people.

Other than in New York, NORCs are not well documented in other U.S. states or other countries. However, the U.S. government funded NORC-SSPs in 26 states between 2002 and 2010. From 2002 to 2008, The Jewish Federations of North America (JFNA) helped Jewish federations and their beneficiary agencies to secure federal demonstration grants in 45 communities with NORC-SSPs. In some parts of Canada, the government developed NORC-SSPs based on the experience of the U.S., but they call the program Oasis.

## 3. Materials and Methods

The research presented here employs the content-analysis-based review method [40] and aims to review the literature systematically and rationalise the outcome of NORC evolvement. Scopus and Thomson ISI Web of Science (WoS), the abstract and citation database of peer-reviewed literature, were used to select journal articles that are related to naturally occurring retirement communities (NORCs) published in authoritative and well-acknowledged scholarly journals in selected areas such as the built environment, gerontology, social work, public health, psychology, safety, etc.

The procedure for searching for relevant literature and then reviewing, selecting, and analysing it is depicted in Figure 1. Planning was performed in the central collection of Scopus and WoS using the keywords "naturally occurring retirement community", "naturally occurring retirement communities", "NORC", "NORCs", "NORC supportive service program", and "NORC-SSP" appearing in the title, abstract, and keyword section of records. The result of the preliminary search provided an initial number (n = 87) of publications that related to NORCs. The abstracts of the initial results were reviewed to screen for the articles most relevant to the topic. The articles that only mentioned NORCs for comparison with other studies but did not offer substantive conclusions related to the topic were removed. After investigating the abstracts and topics of the 87 articles, a total of 49 articles related to the topic and focused exclusively on NORCs were found, with publication dates from 1900 to 2021. Once the relevant articles were determined (n = 49), we began the time-consuming work of highlighting the objectives, methodologies, and findings of each article, all of which needed to be read in detail and potentially more than once when necessary.

In the data analysis stage, NVivo was used to categorise themes and findings, following a top-down approach associated with three typical content analysis methods, i.e., conventional, directed, and summative, in order to interpret the meaning in the content of the textual data [41]. The criteria used in this paper to categorise the findings of the selected articles align with the thematic classification method. Specifically, the categorisation of findings follows a more detailed approach to analysing the multiple findings, innovations, and discoveries generated by each research project. The results present a review of the

research themes and findings consolidated as "NORC Program Development and Assessment", "NORC Living Environment", and "Wellbeing of Older Residents in NORCs", as shown in Table 2.

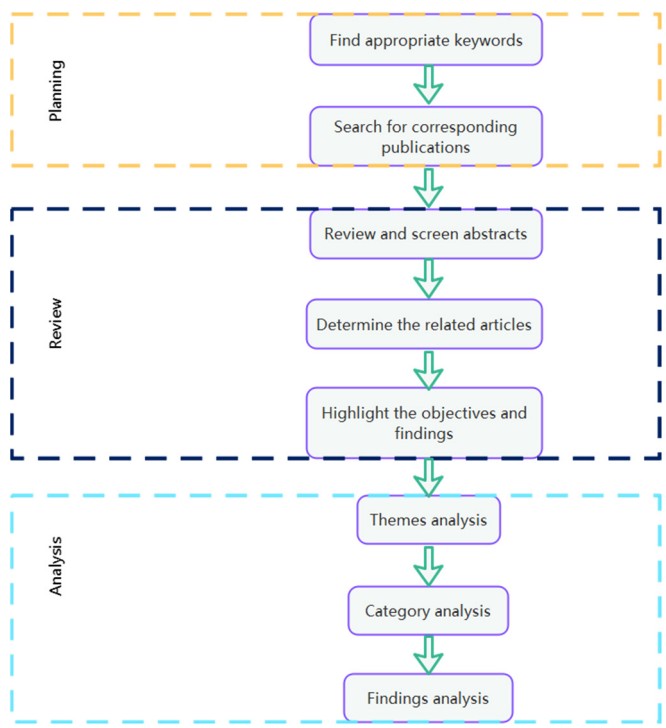

**Figure 1.** Flowchart of implementation process.

## 4. Results

### 4.1. Publication Profile

As shown in Figure 2, since the emergence of the first NORC publication in 1986, the number of yearly publications remained at around 1.5 and no noticeable increase was observed until 2010. The year 2010 witnessed a significant increase in publications, with 17 journal articles concerning NORCs published that year. However, the number of publications per year has since returned to the pre-2010 level with only a slight increase.

Table 2 shows that the leading author was Emily A. Greenfield, who published seven articles (n = 7) about NORCs as first author between 2012 and 2017. Given that NORCs represent a new concept in ageing in place, it is not surprising that the majority of articles were published in journals with a focus on ageing and older people. About one-third of these articles were published in the *Journal of Housing for the Elderly* (n = 16), which has a focus on ageing and housing issues, and these papers are mainly from a policy, planning, and governmental perspective. The *Journal of Gerontological Social Work*, *the Journal of Applied Gerontology*, *The Gerontologist*, and *the Journal of Community Practice* each published three articles.

Regarding research methods, twenty-seven papers adopted qualitative methods such as interviews, focus groups, and case studies (n = 27); fourteen papers used quantitative methods such as questionnaires, surveys, and census data; and seven used a mixed method. Given that the concept of NORCs emerged in the U.S., most of the study regions were in North America (n = 47), including 43 in the U.S. and 3 in Canada, and only two articles each targeted Australia and South Korea.

**Table 2.** NORC Publication Theme Categories.

| Theme | Category | Year | Author | Journal | Title | Methodology | Region |
|---|---|---|---|---|---|---|---|
| NORC Program Development and Assessment | Program Model Development | 2016 | Greenfield and Frantz [42] | *Journal of Community Practice* | Sustainability Processes among Naturally Occurring Retirement Community Supportive Service Programs | Questionnaire, survey, and interview (n = 53) and Thematic analysis | U.S. |
| | | 2013 | Greenfield [43] | *Journal of Housing for the Elderly* | The Longevity of Community Aging Initiatives: A Framework for Describing NORC Programs' Sustainability Goals and Strategies | Interview (n = 10) and grounded theory | U.S. |
| | | 2010 | Kloseck and Crilly [44] | *Journal of Housing for the Elderly* | Naturally Occurring Retirement Communities: Untapped Resources to Enable Optimal Aging at Home | Partnership model | Canada |
| | | 2011 | Guo and Castillo [26] | *Ageing International* | The U.S. Long Term Care System: Development and Expansion of Naturally Occurring Retirement Communities as an Innovative Model for Aging in Place | Evidence-based analysis | U.S. |
| | | 2014 | Greenfield [45] | *Journal of Applied Gerontology* | Community Aging Initiatives and Social Capital: Developing Theories of Change in the Context of NORC Supportive Service Programs | Interview (n = 10) and grounded theory | U.S. |
| | | 2010 | Bedney and Goldberg [9] | *Journal of Housing for the Elderly* | Aging in Place in Naturally Occurring Retirement Communities: Transforming Aging Through Supportive Service Programs | Case study | U.S. |
| | NORC Assessment | 2010 | Bennett [46] | *Journal of Housing for the Elderly* | Exploration and Assessment of the NORC Transformation Process | Interview (n = 49) and historical document analysis | U.S. |
| | | 2007 | Maclaren and Landsberg [35] | *Journal of Gerontological Social Work* | History, Accomplishments, Issues and Prospects of Supportive Service Programs in Naturally Occurring Retirement Communities in New York State: Lessons Learned | Case study | U.S. |
| | | 2002 | Pine and Pine [47] | *Journal of Aging & Social Policy* | Naturally Occurring Retirement Community-Supportive Service Program | Policy analysis | U.S. |
| | | 1999 | Bassuk [48] | *Care Management Journals* | NORC Supportive Service Programs: Effective and Innovative Programs That Support Seniors Living in the Community | Case study | U.S. |

**Table 2.** *Cont.*

| Theme | Category | Year | Author | Journal | Title | Methodology | Region |
|---|---|---|---|---|---|---|---|
| | | 2010 | Cohen-Mansfield and Dakheel-Ali [49] | *Health Promotion International* | The Impact of a Naturally Occurring Retirement Communities Service Program in Maryland, USA | Questionnaire survey (n = 128), *t*-tests, and X2 analyses | U.S. |
| | | 2009 | Anetzberger [50] | *Clinical Gerontologist* | Community Options of Greater Cleveland, Ohio: Preliminary Evaluation of a Naturally Occurring Retirement Community Program | Questionnaire survey (n = 191), descriptive data analysis, frequency counts, percentage distributions, measures of central tendency, and content analysis | U.S. |
| | | 1990 | Hunt and Ross [27] | *The Gerontologist* | Naturally Occurring Retirement Communities: A Multiattribute Examination of Desirability Factors | Interview (n = 143) and MAUT scaling procedure | U.S. |
| | | 2010 | Elbert and Neufeld [51] | *Journal of Housing for the Elderly* | Indicators of a Successful Naturally Occurring Retirement Community: A Case Study | Case study | U.S. |
| | | 2021 | Park and Choi [52] | *Sustainability* | Factors Affecting the Intention of Multi-Family House Residents to Age in Place in a Potential Naturally Occurring Retirement Community of Seoul in South Korea | Questionnaire survey (n = 289), descriptive statistics, correlation tests, *t*-test, factor analysis, and regression analysis by SPSS | Korea |
| | | 1999 | Marshall and Hunt [53] | *Journal of Housing for the Elderly* | Rural Naturally Occurring Retirement Communities: A Community Assessment Procedure | U.S. Census survey and stepwise discriminant analysis | U.S. |
| | | 2010 | Bronstein and Kenaley [54] | *Journal of Housing for the Elderly* | Learning from Vertical NORCs: Challenges and Recommendations for Horizontal NORCs | Review and comparison | U.S. |
| | | 1986 | Hunt and Gunter-Hunt [25] | *Journal of Housing for the Elderly* | Naturally Occurring Retirement Communities | Case study | U.S. |
| | | 2021 | E and Xia [34] | *Sustainability* | Sustainable Urban Development for Older Australians: Understanding the Formation of Naturally Occurring Retirement Communities in the Greater Brisbane Region | Australia census survey, Global Moran's I, and Local Moran's I | Australia |

**Table 2.** *Cont.*

| Theme | Category | Year | Author | Journal | Title | Methodology | Region |
|---|---|---|---|---|---|---|---|
| | | 2015 | Rivera-Hernandez and Yamashita [21] | *Journals of Gerontology Series B: Psychological Sciences and Social Sciences* | Identifying Naturally Occurring Retirement Communities: A Spatial Analysis | U.S. Census survey, Global Moran's I, and Local Moran's I | U.S. |
| | | 2012 | Greenfield and Scharlach [55] | *Journal of Aging Studies* | A Conceptual Framework for Examining the Promise of the NORC Program and Village Models to Promote Aging in Place | Conceptual model | U.S. |
| | NORC Challenges | 2007 | Carpenter and Edwards [56] | *Journal of Gerontological Social Work* | Anticipating Relocation: Concerns About Moving Among NORC Residents | Interview (n = 324), iterative stepwise approach, Chi-square and *t*-test, logistic regression | U.S. |
| | | 2017 | Davitt and Greenfield [57] | *Journal of Community Practice* | Challenges to Engaging Diverse Participants in Community-Based Aging in Place Initiatives | National survey, descriptive statistics, Scheffe post hoc test, Pearson, cross-tabulation procedures, and content analysis | U.S. |
| | | 2010 | Lun [58] | *Journal of Religion & Spirituality in Social Work: Social Thought* | The Correlate of Religion Involvement and Formal Service Use Among Community-Dwelling Elders: An Explorative Case of Naturally Occurring Retirement Community | Interview (n = 521), hierarchical logistic regression analyses, and Andersen-Newman behavioural model | U.S. |
| | | 2010 | Enguidanos and Pynoos [59] | *Journal of Housing for the Elderly* | Comparison of Barriers and Facilitators in Developing NORC Programs: A Tale of Two Communities | Direct observation and in-depth interviews (n = 613) | U.S. |
| | | 2010 | Vladeck and Segel [38] | *Journal of Housing for the Elderly* | Identifying Risks to Healthy Aging in New York City's Varied NORCs | 75-item survey instrument | U.S. |

**Table 2.** *Cont.*

| Theme | Category | Year | Author | Journal | Title | Methodology | Region |
|-------|----------|------|--------|---------|-------|-------------|--------|
| NORC Living Environment | Social Environment | 2016 | Greenfield [60] | *The Gerontologist* | Support from Neighbors and Aging in Place: Can NORC Programs Make a Difference? | Interview (n = 41) and grounded theory | U.S. |
| | | 2014 | Ivery [61] | *Journal of Community Practice* | The NORC Supportive Services Model: The Role of Social Capital in Community Aging Initiatives | Interview (n = 282) and descriptive statistics | U.S. |
| | | 2011 | Bronstein and Gellis [62] | *Journal of Applied Gerontology* | A Neighborhood Naturally Occurring Retirement Community: Views From Providers and Residents | Interview and theme analysis | U.S. |
| | | 2010 | Ivery and Akstein-Kahan [8] | *Journal of Gerontological Social Work* | NORC Supportive Services Model Implementation and Community Capacity | Interview (n = 24) and open coding process | U.S. |
| | | 2017 | Greenfield and Mauldin [63] | *Ageing & Society* | Participation in Community Activities through Naturally Occurring Retirement Community (NORC) Supportive Service Programs | Interview (n = 41) and grounded theory | U.S. |
| | | 2015 | Greenfield and Fedor [64] | *Journal of Gerontological Social Work* | Characterizing Older Adults' Involvement in Naturally Occurring Retirement Community (NORC) Supportive Service Programs | Interview (n = 35) and progressive, multiphased coding process | U.S. |
| | | 2010 | Ivery and Akstein-Kahan [6] | *Administration in Social Work* | The Naturally Occurring Retirement Community (NORC) Initiative in Georgia: Developing and Managing Collaborative Partnerships to Support Older Adults | Focus groups (n = 500) and interview (n = 150) | U.S. |
| | | 2010 | Susan and Jon [65] | *Cityscape (Washington, D.C.)* | Integrating Community Services Within a NORC: The Park La Brea Experience | Case study | U.S. |
| | | 2011 | Lun [66] | *Journal of Social Service Research* | Exploring Formal Service Use by Older Chinese: A Case Study on a Naturally Occurring Retirement Community | Interview (n = 296) and regression analysis | U.S. |
| | Built Environment | 2010 | Tremoulet [67] | *Journal of Housing for the Elderly* | Manufactured Home Parks: NORCs Awaiting Discovery | Focus group (n = 48) | U.S. |
| | | 2014 | Aurand and Miles [68] | *Journal of Housing for the Elderly* | Local Environment of Neighborhood Naturally Occurring Retirement Communities (NORCs) in a Mid-Sized U.S. City | U.S. census survey and descriptive statistics | U.S. |

**Table 2.** *Cont.*

| Theme | Category | Year | Author | Journal | Title | Methodology | Region |
|---|---|---|---|---|---|---|---|
| Wellbeing of Older Residents in NORCs | Security | 2014 | Kloseck and Gutman [69] | *Journal of Housing for the Elderly* | Naturally Occurring Retirement Community (NORC) Residents Have a False Sense of Security That Could Jeopardize Their Safety in a Disaster | Peer-led focus group (n = 12) | Canada |
| | Health | 2016 | McClive-Reed and Gellis [70] | *Journal of Gerontological Social Work* | Psychological Distress and Help-Seeking by Residents of a Neighborhood Naturally Occurring Retirement Community (NNORC) | Questionnaire survey (n = 226), linear regression model, logic regression model, best-fitting model and proportional odds model | U.S. |
| | | 2020 | Chippendale [71] | *Health Education & Behavior* | Outdoor Falls Prevention Strategy Use and Neighborhood Walkability Among Naturally Occurring Retirement Community Residents | Questionnaire survey (n = 97), descriptive analysis, and Chi-square test | U.S. |
| | | 2009 | Pickard and Fengyan [72] | *Research on Aging* | Older Adults Seeking Mental Health Counseling in a NORC | Questionnaire survey (n = 317) and multinomial logistic regression | U.S. |
| | | 2013 | Dale and Renee [73] | *International Public Health Journal* | Implementing and Disseminating a Fall Prevention Program in At-risk Older Adults Living in a Naturally Occurring Retirement Community-Supportive Services Program | Single group pre-post survey (n = 93), HFRA tool, and McNemar's test | U.S. |
| | | 2006 | Masotti and Fick [74] | *American Journal of Public Health* | Healthy Naturally Occurring Retirement Communities: A Low-Cost Approach to Facilitating Healthy Aging | Health model | Canada |
| | | 2010 | Masotti and Fick [75] | *Journal of Housing for the Elderly* | Healthy Naturally Occurring Retirement Communities: The Need for Increased Collaboration Between Local Public Health Agencies and Municipal Government | Discursive analysis | Canada |

**Table 2.** *Cont.*

| Theme | Category | Year | Author | Journal | Title | Methodology | Region |
|---|---|---|---|---|---|---|---|
| | Active Living | 2010 | Grant-Savela [76] | *Journal of Housing for the Elderly* | The Influence of Self-Selection on Older Adults' Active Living in a Naturally Occurring Retirement Community | Questionnaire survey (n = 197), Chi-square test, Mann–Whitney U test, Kruskal–Wallis test, and Spearman's rank-order correlation | U.S. |
| | | 2009 | Hildebrand and Neufeld [77] | *The Gerontologist* | Recruiting Older Adults Into a Physical Activity Promotion Program: Active Living Every Day Offered in a Naturally Occurring Retirement Community | Interview and transtheoretical model (TTM) (n = 50) | U.S. |
| | | 2010 | Ahrentzen [78] | *Journal of Housing for the Elderly* | On Their Own Turf: Community Design and Active Aging in a Naturally Occurring Retirement Community | Questionnaire survey (n = 719) and descriptive statistics | U.S. |
| | | 2007 | Carpenter and Buday [79] | *Computers in Human Behavior* | Computer Use Among Older Adults in a Naturally Occurring Retirement Community | Questionnaire survey (n = 324), descriptive statistics, and *t*-test | U.S. |
| | | 2010 | Grant-Savela [80] | *Journal of Applied Gerontology* | Active Living Among Older Residents of a Rural Naturally Occurring Retirement Community | Questionnaire survey (n = 197), descriptive statistics, Chi-square tests, Mann–Whitney U, multivariate analyses, and Spearman's rank-order correlations, etc. | U.S. |

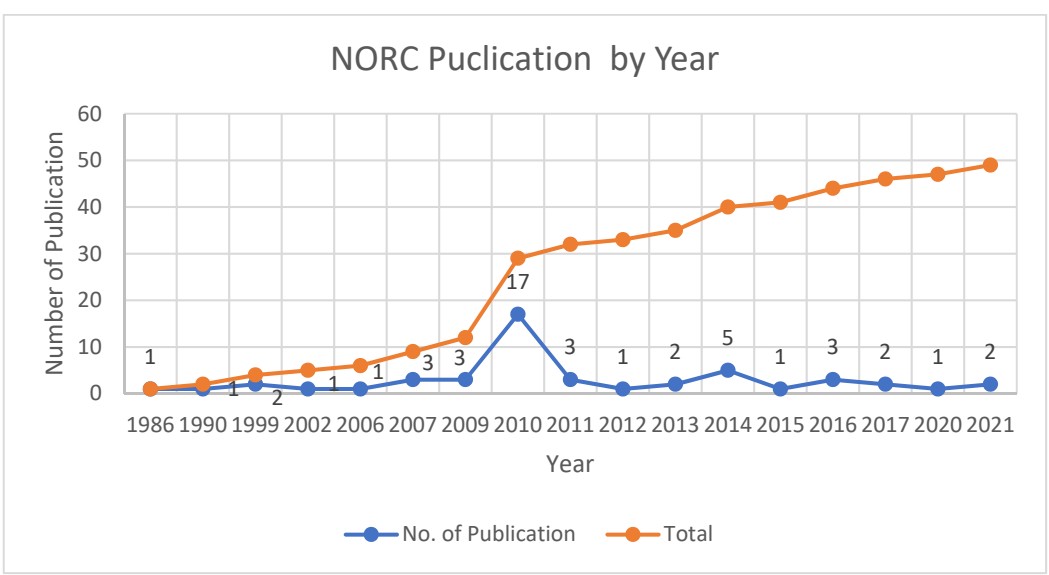

**Figure 2.** Number of NORC Papers Published by Year.

*4.2. Research Themes and Findings*

Figure 3 presents the three-level classification of findings with the number of articles published with respect to the finding categories. Under the theme of "NORC Program Development and Assessment" are included the subcategories of NORC program development (n = 6), NORC assessment (n = 15), and NORC challenges (n = 5), for a total of 26 articles regarding this theme. Likewise, "NORC Living Environment" contains the social environment (n = 9) and built environment (n = 2) categories. "Wellbeing of Older Residents in NORCs" includes security (n = 1), health (n = 6), and active living (n = 5), representing 12 articles out of the 49 total publications.

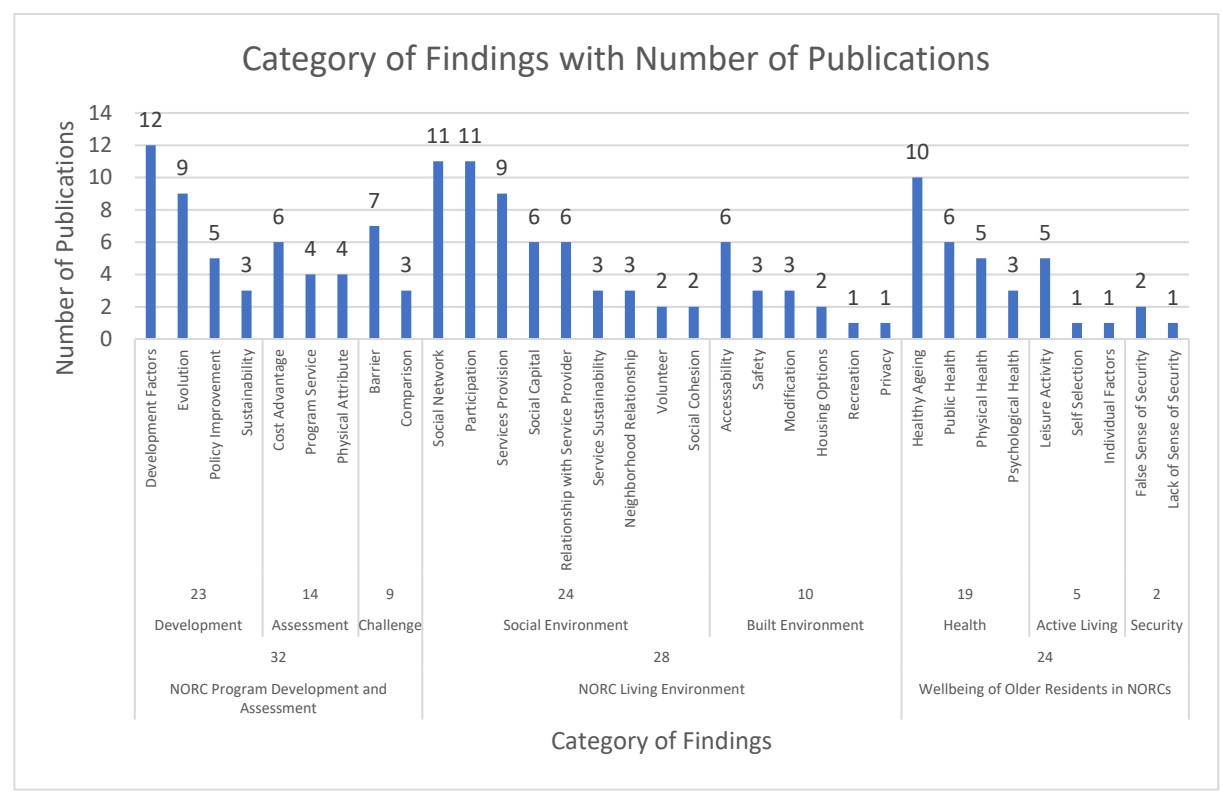

**Figure 3.** The categories of findings with the number of publications for each.

It should be pointed out that, although some topics are less frequently discussed (e.g., there is only one paper related to security among NORC residents), it does not necessarily mean they are unimportant. On the contrary, these may be areas for future research.

### 4.2.1. NORC Development and Assessment

NORC Development

As shown in Figure 3, NORC development (i.e., how to successfully develop and operate NORCs) is the most frequently discussed research category (with 23 papers), among which the "development factor" was most examined with 12 articles. The factors discussed all provided feasible suggestions for NORC development from different perspectives. Ivery and Akstein-Kahan [6] found that the formation of NORCs was influenced by several factors, with an understanding of the organisational capacity of the community being critical, as well as the need to strategically select partners and NORC sites that optimise capitalisation of priority status, and the need for marketing items developed by partnerships to increase visibility, etc. Meanwhile, Guo and Castillo [26] argued that partnerships play a crucial role in ensuring NORC programs succeed by bringing together professional services, both human and technics, in the community for the beneficial exploitation of resources to meet the social, emotional, health, physical, and environmental needs of the communities and their older adults. Enguidanos and Pynoos [59] elucidated that providing a flexible mix of services, obtaining start-up funds, and engaging powerful partners were factors that facilitated supportive service program adoption in both horizontal and vertical NORCs.

Furthermore, the findings in the development category pertain to policy improvement (n = 5), evolution (n = 9), and sustainability (n = 3) in addition to the discovery of development factors. E and Xia [34] discussed policy development and suggested that researchers, communities, and local governments could work together to identify NORCs and their formation patterns so as to better understand the economic activities, local history, and experiences of older adults in NORCs to improve policy planning and development. However, Masotti and Fick [74] investigated the evolvement of healthy NORCs and concluded that when the elderly population grows and its political and market weight increases, a NORC may become a healthy NORC. Regarding the sustainability of NORCs, researchers found several factors that support a sustainable strategy, including program effectiveness, the role of the coordinator, inter-organisational partnerships, and sufficient financial resources, among others [42,43,46].

NORC Assessment

A total of 14 articles discussed NORC assessment (i.e., assessing the features of NORCs); these may be further categorised as cost advantages, program services, and physical attributes. Of these, six articles assessed the cost advantages of NORCs, and they generally concluded that NORCs have cost advantages compared to other forms of retirement communities. Masotti and Fick [74] claimed that the healthy NORC policies they suggested can facilitate healthy ageing with lower costs compared with health care services. Elbert and Neufeld [51] seem to agree with the above statement; they suggested that NORCs had proven a truly cost-effective mechanism for supporting older adults' ageing in place, as well as for enabling communities to maintain their vibrancy, diversity, and health. Chaikin and Pekmezaris [73] highlighted, from a health perspective, that as a result of the trusting relationship that develops between NORC staff and residents and the large potential reach of the NORC-SSP model, NORC-SSPs may prove to be a cost-effective, valuable resource for the prevention and treatment of chronic conditions and diseases among older adults living in the community.

NORC program services and physical attributes are covered by four articles each. Program services were assessed as follows: they were more likely to be used by older Chinese women [66], they were better able to serve older adults with increased funding and other resources [26], they offered concrete supports that were important, such as health and mental health care [62], and they required client acceptability, which played a critical role

in NORC services for older residents [58]. For physical attributes, it was found that vertical NORCs were more successful [59], several NORCs were located in the most populous urban areas, and older populations were more likely to live close together [21].

NORC Challenges

The reasons for older adults' unwillingness to accept NORCs are numerous, including a reluctance to trust, an unwillingness to relocate, and non-engagement with the activities. NORC barriers also refer to development difficulties and ownership attribution. Comparisons of vertical and horizontal NORCs, older residents in NORCs and non-NORCs, and older and younger NORC residents are all NORC challenges linked to social isolation, service delivery, housing environment, and living convenience for both service providers and older residents.

McClive-Reed and Gellis [70] suggested a possible link between help-seeking and powerful others, as residents were more likely to seek help if they believed that powerful others were primarily responsible for keeping them well; however, despite demonstrated needs, low levels of help-seeking were reported. Carpenter and Edwards [56] researched relocation and underscored that changes in residents' health status were the most common cause of concern about moving, as well as concerns about available financial resources. Psychologically, residents who were concerned about moving appear more depressed than those who did not. Tremoulet [67] also discussed the issues related to manufactured houses in NORCs, and one factor was the degree to which parks were open to outsiders; another was their willingness to allow services to be delivered in their parks, particularly if they were delivered in group settings. Enguidanos and Pynoos [59] compared the barriers and facilitators to developing NORC programs and highlighted that a number of challenges were encountered, including building senior empowerment quickly as well as planning for long-term sustainability and creating customised service packages.

4.2.2. NORC Living Environment

Social Environment

The social environment of NORCs has received considerable attention from scholars, with 24 of the 49 articles discussing the link between the social environment and NORCs. The social environment discussed includes the roles of volunteers, social network, social cohesion, social capital, service provision, relationship with service providers, participation, and neighbourhood relationships.

Eleven articles concluded that participation in the NORC supportive service program generates positive outcomes for older adults. Greenfield and Fedor [64] indicated that participants who were highly involved in the NORC program were either active leaders in the program or relied heavily on the program for assistance and socialisation. Meanwhile, they also discussed that a low level of involvement was indicative of a lack of a positive relationship between the older adults and the programs. It has been reported in some cases that participants deliberately distanced themselves from the NORC programs or failed to recognise that they were taking advantage of the program. Greenfield and Mauldin [63] illustrated that relationships with staff were cited as an important factor influencing participation. This indicated the importance of training, hiring, and supporting staff who were qualified to deliver specific types of services as well as engage older adults more broadly in the NORC program. In their study, the authors also outlined five factors significantly affecting participation in activities, including the need for additional socialisation, health conditions, relationships with staff, attractiveness of the activities, and the perspectives of other participants.

Another primary finding regarding the social environment was related to social networks. It is generally accepted that most of these activities provided by NORC programs have the possibility to improve social networks and reduce isolation. Greenfield [45] suggested facilitating the interaction between older adults via community programs and individual services in order to improve their wellbeing and strengthen informal networks

of support. As part of NORC activities, older adults have the opportunity to collaborate on common goals and find mutual support as they adopt healthier behaviours and lifestyles. Ahrentzen [78] discovered it was possible that the length of time that people had lived in their community, their interactions with their neighbours, and even the large number of other older individuals that lived there contributed to the sense of neighbourhood cohesion and belonging. Ivery [61] highlighted that despite the fact that NORC programs were not created to change the environment, the services provided could be leveraged to develop the social capital that is needed to modify both the physical and social environment to meet the needs of older people.

It is also important to note that volunteers, social cohesion, and neighbourhood relationship, which were found in the study but did not receive much attention from many authors, are also essential components of the NORC social environment. Greenfield and Frantz [42] underscored that engaging volunteers had been identified as a strategy to enhance sustainability by a number of respondents. Park and Choi [52] also explained that the reasons for intending to move to the neighbourhood were related to the physical environment and the relationships with people in the neighbourhood.

Built Environment

In order to encourage independence and physical activity, the built environment can be a way to assist older people in accessing local services or providing better connections to public transportation so they can reach their destinations and recreation centres easily. Ten papers refer to the built environment on the topics of safety, recreation, privacy, modification, housing options, and accessibility. They mainly discussed safety as being generally a key factor in active ageing, with in-migrants self-selecting into the NORC because of recreation and modifications to physical features of the environment to improve active ageing (features that benefit a larger population than just older people), and they highlighted that proximity to amenities was a key factor in walking behaviour, etc.

Hunt and Gunter-Hunt [25] stressed that it was important to consider built environment factors, such as the degree of safety and proximity to peers, that influence the level of desirability of a living arrangement. Tremoulet [67] promoted manufactured housing and highlighted that, compared to traditional neighbourhoods with single-family detached homes or apartment buildings, manufactured home parks might provide more privacy and more flexibility. Grant-Savela [76] emphasised the fact that NORCs attracted immigrants due to the opportunities for outdoor recreation, the ability to participate in specific recreational activities, and the opportunity to promote an active lifestyle. However, Aurand and Miles [68] found that, in some areas, even a NORC located in an urban district might not have access to a variety of places favoured by older people, such as grocery stores, restaurants, post offices, libraries, churches, or a community centre.

4.2.3. Wellbeing of Older Residents in NORCs
Health

Nineteen articles investigated the health of residents in NORCs, covering healthy ageing, public health, psychological health, and physical health. Health is the most concerning topic for older adults who would like to age at home. Healthy ageing includes topics such as NORC-SSP promoting healthy ageing in place in terms of social contacts, service provision, participant involvement, resource offerings, etc. Meanwhile, public health discusses the health risks, preventing entry into nursing homes and hospitals, prolonging independent living, changing perceptions of ageing, etc. Psychological health refers to topics such as the stress of seeking help, the fact that more than half of residents reported loneliness and attendant crises in mental health services, etc. Research on physical health is mainly related to fall prevention, chronic illnesses, pain, etc.

Pickard and Fengyan [72] articulated that the most frequent attendees of religious services asked religious leaders for help more often than they asked other formal sources, but they were also less likely to ask for help from other formal sources than to not ask for

anything. McClive-Reed and Gellis [70] illustrated that 21.6% of 268 residents reported depressive feelings, but only 8.6% of those with elevated scores reported receiving counselling or mental health services. Chippendale [71] worked on an outdoor fall-prevention strategy and showcased that, while some outdoor fall-prevention strategies were consistently utilised by participants, including looking ahead and holding railings on stairs, other strategies were less frequently employed, such as carrying fewer items, not hurrying, and wearing appropriate footwear. McClive-Reed and Gellis [70] explained that NNORC residents were primarily affected by chronic illnesses and pain, and many also experienced psychological distress and loneliness. Bedney and Goldberg [9] explored the fact that NORC programs facilitated healthy ageing in two ways: firstly, through collaboration among civic organisations, social service agencies, and building owners, older adults were offered programs and services in a coordinated, systematic manner; secondly, and most importantly, older adults were involved in the decision-making process.

Active Living in NORCs

Active living has been accepted as a better approach to healthy ageing. Five articles examined the active living of older residents in NORCs. The topic of self-selection discussed the motivations of in-migrants to NORCs. The leisure activities topic underscored how to best encourage activities such as walking to support active living among older adults. The individual factors topic explored some individual factors most likely influencing the patterns and types of active living.

Grant-Savela [76] claimed that when self-selection was considered instead of immigration alone, three socio-physical characteristics were more pronounced. Firstly, it was shown that self-selectors were more likely than other in-migrants or long-time residents to walk in order to enjoy access to fresh air. Furthermore, there is also a strong possibility that active living attracted spouses or at least encouraged them to support their spouse's decision. In addition, participants who recently moved demonstrated a desire for social networks by walking or riding bicycles to meet people. Grant-Savela [80] also explained in another article that NORC socio-physical characteristics were more closely related to leisure activities than to household activities, which might be affected by individual factors. Aurand and Miles [68] argued that neighbourhoods with NORCs, particularly those with older adults, were helpful for such initiatives, as older adults have become increasingly interested in how to live an active lifestyle.

Security

Two articles about security look into the lack of a sense of security and a false sense of security in NORCs. Kloseck and Gutman [69] mentioned that, in most cases, seniors did not consider the need for emergency planning or take steps to be prepared in case of an emergency. They also illustrated that mobility is crucial during evacuations, and the elderly were unsure how they would evacuate from high-rise apartment buildings without elevators; meanwhile, seniors were unlikely to seek out emergency planning information on their own. Concerning a false sense of security, Kloseck and Gutman [69] also discussed that seniors always had the false sense that they would be able to handle any emergency situation that might arise and did not need to prepare. In general, seniors assumed that the city and the management of their apartment complex would provide all the information and emergency support they needed, although few were aware of how the information would be provided or what emergency assistance would be available.

## 5. Discussion and Future Research Directions

This research conducted an overview of naturally occurring retirement communities (NORCs), covering the definition, history, evolvement, NORC supportive service programs, development, assessment, the living environment, and the wellbeing of older residents. This research aimed to summarise previous research findings in a comprehensive manner in order to explore the impact of NORCs on ageing in place and to guide future research.

Various subjects were examined, including development, assessment, challenges, the social environment, the built environment, security, health, and active living; these topics encompass the pros and cons of healthy ageing among older adults in NORCs.

Generally, NORC programs have been deemed by most of the prior research as an effective way to encourage healthy ageing so that older adults can successfully age in place. As evident in the study of NORC promotion, local governments and service providers have a direct influence on the acceptance of the program by the elderly, since political leaders know best the history, characteristics, and needs of older adults [26]. Specifically, the government should evaluate policies that address business and residential zoning, public health safety, and access to health care. Furthermore, it is important to evaluate the political and economic feasibility of facilitating NORCs in different geographical and socioeconomic settings. Consequently, NORCs have time-sensitive and comprehensive policy implications because the health of older people is dependent on a comprehensive understanding of community services. On the other hand, all stakeholders, from businesses to health professionals, should contribute their insight, understanding, and collaboration [44]. While the community can partially rely on itself, it requires infusions of energy to survive. These infusions originate from outside and are often provided by professionals. Thus, it is imperative to develop a close relationship and collaborative partnership between NORC residents and service providers in order to facilitate the healthy development of NORCs.

It has been shown that the social environment plays a significant role in promoting the wellbeing of older adults in their ageing in place. Firstly, NORC programs offer participants a variety of opportunities to build and expand their social networks. Members of the NORC-SSP may benefit from group activities offered by the organisation, including interest groups and cultural and educational events, which may facilitate closer relationships [55]. As a result of continuing growth in membership, NORC-SSP has developed a significant range of services and programs, and community members surveyed believed that NORC-SSP positively affected their community [60]. It was a major success of the NORC-SSP to help older adults connect with their community and develop social networks. Furthermore, NORC-SSPs have the potential to serve as a mechanism by which social capital can be generated, since they can simultaneously serve as a means for seniors to engage in civic life, as well as a means by which members of the larger community may work collaboratively with older adults to enhance community life and function [61]. Moreover, NORC-SSP offers activities and services for older adults utilising resources such as funding, staff, inter-organisational partnerships, volunteers, etc. Given that the NORC-SSP is grounded in the core principles of being responsive to the community and maximising consumer participation among older adults, the participation of older adults is perceived as having an impact on the resources available to provide programs and services, which is the underlying motivation for NORC development. Volunteer recruitment and retention, however, were constantly challenging. In spite of this, the NORC-SSP managed to engage a substantial number of volunteers who benefited from their participation and contributed to the wellbeing of others, which is also part of the NORC long-term strategy for sustainable development.

According to Masotti and Fick [74], when the built environment meets the physical needs of older people, they will remain in or move to that environment, which will then shape the NORC into a healthy NORC. Many studies examining the built environment place a high priority on safety. Heavy traffic, poor sidewalks, inadequate street lighting, unattended dogs, and undesirable kids or strangers all contribute to safety concerns [78]. Hunt and Gunter-Hunt [25] emphasised the importance of safety and peers close to the same age when they first presented the concept of a NORC. Moreover, the built environment confers benefits beyond older adults; as Aurand and Miles [68] highlighted, physical modifications to a neighbourhood for active ageing often benefited a wider population than just the elderly, and sidewalk improvements enhanced the quality of life for people of all ages. Addy and Wilson [81] also stressed that it was most common to see physical

activity on neighbourhood streets due to their convenience and proximity to homes as well as the opportunity to interact with neighbours.

In spite of the fact that the NORC literature addresses the aspects of development, the wellbeing of older residents, and the living environment, there are still many areas that need further exploration. Firstly, there is little systematic evaluation of the built environment in NORCs, and the current research is confined to a few areas, which means that policy makers and community environment planners are uncertain of the deterministic role of the built environment in influencing older adults' wellbeing and independent living in NORCs. In other words, the question of which factors of the built environment may affect older people staying in NORCs and facilitate ageing in place need to be investigated, as does the question of how these factors influence these issues.

Secondly, it appears that research on NORCs has been stagnant in recent years, as the number of articles published after 2010 is not significantly greater than that before 2010. However, given that NORCs have become a factual phenomenon, and a growing number of communities, suburbs, and towns are meeting the criteria for NORCs due to their increasing elderly populations, it is essential to increase research resources focused on NORCs and to intensify the research into service programs in order to meet the challenges of global ageing.

Thirdly, although the phenomenon of NORC formation already exists in other regions of the world, NORC support service programs are currently only offered in North America. No other regions of the world provide corresponding systematic services in accordance with NORC distribution. It is important for policy makers in other countries to recognise the advantages of NORC supportive service programs in integrating social resources, improving the efficiency of resource utilisation, saving public healthcare investment, and enhancing the physical and mental health of older people. Globalising NORC research may provide a viable initiative for addressing global ageing and promoting ageing in place.

Fourthly, to date, there are a few publicly available sources of information about NORCs. New York State is the only local government that provides detailed information about NORC programs on its website. NORC-related information should be consolidated and published from a social research perspective, which would be statistically significant for future research on NORCs and their service programs.

Last but not least, with the rapid advancement of computer technology in recent years, statistical research methods related to social computing science can be more intelligently studied with the help of artificial intelligence in order to study the life patterns of large sample groups and the refined needs of individuals. Future research on NORCs and the optimal allocation of resources related to NORC services can be pinpointed with the help of artificial intelligence algorithms to eventually guide policy making.

## 6. Conclusions

This paper presented an overview of publications about naturally occurring retirement communities and their supportive service programs. It closely examined whether NORCs support ageing in place by examining the areas of NORC program development and assessment, NORC living environment, and the wellbeing of older residents in NORCs. The results show that the research areas concerning NORC development and the social environment of NORCs were most explored. It is widely agreed that NORCs play a positive role in facilitating ageing in place; in particular, the social environment provides the necessary mental support and physical motivation for older adults to live actively. In terms of NORC assessment and the health of older residents in NORCs, topics frequently referred to as well, NORC supportive service programs were determined to be the optimal choice for supporting ageing, both for the individual and for the community from the perspectives of cost, service, and physical attributes. Meanwhile, the topics of physical and psychological health provided a comprehensive examination of older residents, and most of them experienced positive outcomes after participating in the offered programs. In

many studies, challenges such as relocation concerns, trust issues, and ownership problems were identified, and there were calls for solutions in future studies.

This study provided both researchers and stakeholders, such as policymakers, service providers, and older residents, with a useful reference for understanding this ageing alternative. In addition to providing insight into the history and evolvement of NORCs, the paper pointed out potential research topics for future research. The study also has a few limitations. Firstly, information on NORC service programs is rarely made public, which makes it impossible to accurately count the number and distribution of service programs. Moreover, very limited studies on NORCs were published during 2018–2021, probably because of the dramatic changes in the world over the past few years, especially with pandemic outbreaks around the world leading to serious threats to the living environments of older adults. Furthermore, outside of North America, home care programs are identified by different names in other regions, and this study of NORCs focused only on the North American region in the form of NORCs, which makes it challenging to examine the models of ageing in place globally. In addition, the classification of research themes was inevitably subjective, and different methods of classification may lead to different results. However, given that the majority of older people nowadays are unwilling to leave their homes to spend their later life elsewhere, and given that NORC programs provide an alternative way to age at home, it is hoped that this study will serve as a basis for future research on NORCs.

**Author Contributions:** Conceptualization, J.E. and B.X.; methodology, J.E. and B.X.; software, J.E.; validation, B.X., C.S. and Q.C.; formal analysis, J.E.; writing—original draft preparation, J.E.; writing—review and editing, B.X., C.S., Q.C. and X.W.; supervision, B.X. and C.S. All authors have read and agreed to the published version of the manuscript.

**Funding:** This research received no external funding.

**Institutional Review Board Statement:** Not applicable.

**Informed Consent Statement:** Not applicable.

**Data Availability Statement:** Not applicable.

**Conflicts of Interest:** The authors declare no conflict of interest.

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
