# Peer review of "An Overview of Naturally Occurring Retirement Communities (NORCs) for Ageing in Place"

_buildings, doi:10.3390/buildings12050519_

Round 1

Reviewer 1 Report

The paper “An Overview on Naturally Occurring Retirement Communities (NORCs) for Ageing in Place” conducts an overview of NORC, aiming to examine the previous research in a comprehensive manner in order to explore how the NORC impact aging in place and guide future research. The research employs the content analysis of prior NORC related studies to categorize research themes and findings following top-down coding principles. The manuscript is well written and seems to be ready for publication. Though, I see some aspects in the Results section look more like research methods, for example, how you searched and investigated the 87 found publications through Scopus and WoS databases, as well as the research theme explanation.

Author Response

Though, I see some aspects in the Results section look more like research methods, for example, how you searched and investigated the 87 found publications through Scopus and WoS databases, as well as the research theme explanation.

Response: Thank you for your recognition and comments.  We have removed some of the narratives in the Results and added to the Materials and Methods in the line 269-270 and line 277-283 as follows:

“After investigating the abstracts and topics from the 87 articles, a total of 49 articles related to the topic and focused exclusively on NORC were found during the year from 1900 to 2021.”

“The criteria used in this paper to categorize the findings of selected articles align with thematic classification method. Specifically, the categorization of findings describes a more detailed approach of analyzing the multiple findings, innovations, and discoveries generated by each research project.  The results present a review on the research themes and findings consolidated as “NORC Program Development and Assessment”, “NORC Living Environment” and “Wellbeing of Older Residents in NORCs” as shown in Table 2 and Figure 3.”

Reviewer 2 Report

The topics was interesting and relevant to current sitution. The paper was well-written in the first part to provide the background and to identify the importance of the study. The research process was clearly explained. However, in Table 2. NORC Publications Theme Categories, many of the studies were in 2010 or older. Few studies can represent the time period during 2018-2021 or 2022. Will this be the limitation of the study? The reason is because there have been a significant change in the past few years. Several conditions have been changed dramatically and it is highly important to address this point. In addition, most of the studies were from North America, including US and Canada. Will this have the limitation to address the broader scope of Naturally Occurring Retirement Communities in different part of world? The author should address this more clearly. Generally, the presentation of the results was fine. Overall, the paper was well-presented and the topic is highly relevant to today's problem. 

Author Response

  1. The topics was interesting and relevant to current situation. The paper was well-written in the first part to provide the background and to identify the importance of the study. The research process was clearly explained. However, in Table 2. NORC Publications Theme Categories, many of the studies were in 2010 or older. Few studies can represent the time period during 2018-2021 or 2022. Will this be the limitation of the study? The reason is because there have been a significant change in the past few years. Several conditions have been changed dramatically and it is highly important to address this point. 

Response: Thanks for your suggestion and we agree with your point that it is a limitation of the study.  In the Discussion and Future Research Directions, we have highlighted a future research direction of increasing NORC research resources and intensifying the research of its service programs in order to meet the challenges of global ageing.  In the limitation of the study in Conclusion, we have added the point in line 663-666 as follows.

“Moreover, very limited studies on NORC were published during 2018-2021, probably because of the dramatic changes in the world over the past few years, especially with pandemic outbreaks around the world leading to serious threats to the living environment of older adults.”

  1. In addition, most of the studies were from North America, including US and Canada. Will this have the limitation to address the broader scope of Naturally Occurring Retirement Communities in different part of world? The author should address this more clearly. Generally, the presentation of the results was fine. Overall, the paper was well-presented and the topic is highly relevant to today's problem. 

Response: Thanks for your comments. First of all, we agree that it is the limitation of the study (i.e. most of the studies were from North America, including US and Canada), which has been listed in the Conclusion already.  Moreover, in the Discussion and Future Research Directions, we have addressed NORC globalization as a future research direction in the third bullet. In addition, we have added more discussion in line 624-627 as follows:

“Thirdly, although the phenomenon of NORC formation already exist in other regions of the world, NORC support service programs are currently only offered in North America. No other regions of the world provide corresponding systematic services in accordance with NORC distribution. It is important for policy makers in other countries to recognize the advantages of NORC supportive service programs in integrating social resources, improving the efficiency of resource utilization, saving public healthcare investment, and enhancing the physical and mental health of the older people.  Globalizing NORC research may provide a viable initiative for addressing global ageing and promoting ageing in place.”